# Motion Forecasting with Unlikelihood Training

## Abstract

Motion forecasting is essential for making safe and intelligent decisions in robotic applications such as autonomous driving. State-of-the-art methods formulate it as a sequence-to-sequence prediction problem, which is solved in an encoder-decoder framework with a maximum likelihood estimation objective. In this paper, we show that the likelihood objective itself results in a model assigning too much probability to trajectories that are unlikely given the contextual information such as maps and states of surrounding agents. This is despite the fact that many state-of-the-art models do take contextual information as part of their input. We propose a new objective, unlikelihood training, which forces generated trajectories that conflict with contextual information to be assigned a lower probability by our model. We demonstrate that our method can improve state-of-art models' performance on challenging real-world trajectory forecasting datasets (nuScenes and Argoverse) by 8% and reduce the standard deviation by up to 50%. Code will be made available.

## 1 Introduction

For robotic applications deployed in the real world, the ability to foresee the future motions of agents in the surrounding environment plays an essential role for safe and intelligent decision making. This is a very challenging task. For example, in the autonomous driving domain, to predict nearby agents' future trajectories, an agent needs to consider contextual information such as their past trajectories, potential interactions, and maps. State of the art prediction models (Salzmann et al., 2020; Tang & Salakhutdinov, 2019; Rhinehart et al., 2019) directly take contextual information as part of their input and use techniques such as graph neural networks to extract high-level features for prediction. They are typically trained with a maximum likelihood estimation (MLE) objective that maximizes the likelihood of ground truth trajectories in the predicted distribution. Although MLE loss encourages the prediction to be close to the ground truth geometrically, it does not focus on learning a good distribution that is plausible with respect to the contextual information. These models predict trajectories that violate the contextual information (e.g., go to opposite driving direction or out of the driving area) but still closes to ground truth. In contrast, humans can easily notice that these trajectories are unlikely in a specific context. This phenomenon suggests that simply applying MLE loss cannot fully exploit contextual information.

To address the problem, we propose a novel and simple method, unlikelihood training, that injects contextual information into the learning signal. Our loss penalizes the trajectories that violate the contextual information, called negative trajectories, by minimizing their likelihood in the predicted distribution. To generate negative trajectories, we first draw a number of candidate trajectories from our model's predicted distribution. Then, a context checker is used to cut out the trajectories that violate contextual information as negative trajectories. This context checker does not need to be differentiable. By minimizing the likelihood of negative trajectories, the model is forced to use the contextual information to avoid predictions that violate context. Therefore, the prediction quality is improved.

Existing methods (Casas et al., 2020; Park et al., 2020) using contextual information as learning signals either introduce new learning parameters or using high-variance learning methods such as the REINFORCE algorithm (Casas et al., 2020). In contrast, our method injects rich contextual information into the training objective and keeps the training process simple.

Unlikelihood training (Welleck et al., 2019) has been applied to neural text generation. We are the first to propose unlikelihood training for continuous space of trajectories. For the discrete space of token sequences, repeating tokens or n-grams in the generated sequence are chosen as negative tokens. In contrast, we design a context checker to select negative trajectories sampled from the continuous distribution of model predictions.

Our method can be viewed as a simple add-on to any models that estimate the distribution of future trajectories. It improves their performance by encouraging models to focus more on contextual information without increasing the complexity of its original training process.

Our contributions are summarized as follows:

- We propose a novel and simple method, unlikelihood training for motion forecasting in autonomous driving that encourages models to use contextual information by minimizing the likelihood of trajectories that violate contextual information. Our method can be easily incorporated into state-of-the-art models.

- Our experimental results on challenging real-world trajectory forecasting datasets, nuScenes and Argoverse, shows that unlikelihood training can improve prediction performance by 8% and reduce the standard deviation by up to 50%.

## 2 RELATED WORK

In this section, we briefly review the two most related topics.

**Trajectory Forecasting** Trajectory forecasting of dynamic agents, a core problem for robotic applications such as autonomous driving and social robots, has been well studied in the literature. State-of-the-art models solves it as a sequence-to-sequence multi-modal prediction problem (Lee et al., 2017; Cui et al., 2018; Chai et al., 2019; Rhinehart et al., 2019; Kosaraju et al., 2019; Tang & Salakhutdinov, 2019; Ridel et al., 2020; Salzmann et al., 2020; Huang et al., 2019). (Cui et al., 2018; Chai et al., 2019; Ridel et al., 2020) predicts multiple future trajectories without learning low dimensional latent agent behaviors. (Lee et al., 2017; Kosaraju et al., 2019; Rhinehart et al., 2019; Huang et al., 2019) encodes agent behaviors in continuous low dimensional latent space while (Tang & Salakhutdinov, 2019; Salzmann et al., 2020) uses discrete latent variables. Discrete latent variables succinctly capture semantically meaningful modes such as turn left, turn right. (Tang & Salakhutdinov, 2019; Salzmann et al., 2020) learns discrete latent variables without explicit labels. All of them use a maximum likelihood estimation (MLE) objective or its approximations (e.g., VAE). In this paper, we show that MLE loss can ignore contextual information such as maps and states of surrounding agents. As a result, models with such a loss can assign too much probability to unlikely trajectories. We propose an unlikelihood training objective to avoid such cases. All models with the maximum likelihood estimation objective can potentially benefit from our methods.

**Contrastive learning and unlikelihood training** To date, several studies have investigated the possibilities to benefit from negative data. One of the popular direction is contrastive learning. Contrastive learning has achieved significant success in many fields (Oord et al., 2018; Kipf et al., 2019; Ma & Collins, 2018; Abid & Zou, 2019; Welleck et al., 2019). NCE (Ma & Collins, 2018) CPC (Oord et al., 2018) maximizes the mutual information between data and latent representation by a novel contrastive loss to extract useful representation from data. C-SWMs (Kipf et al., 2019) utilizes contrastive learning to learn a better world model for reinforcement learning tasks. Recently, unlikelihood training (Welleck et al., 2019) proposes a new method to utilize negative data. In addition, to maximize the likelihood of the ground truth token, it minimizes the likelihood of negative tokens for better text generation. Their method is on the discrete space of token sequences. Repeating tokens or n-grams in the generated sequence is chosen as negative tokens. In contrast, our proposed method works in the continuous space of trajectories. We design a novel method, context checker, to select negative trajectories.

## 3 METHOD

### 3.1 PROBLEM FORMULATION

We are targeting at better predicting the future trajectory $\mathbf{Y}_i$ of a vehicle $i$ given input $\mathbf{X}_i$. $\mathbf{X}_i$ can include any related information like rasterized maps or past positions of vehicle $i$ and surrounding agents, depends on the design of the method. Here we skip the detailed choice of input and denote it as $\mathbf{X}_i$ for conciseness. Due to different driving strategies, driving intents, and the complex traffic environment, there are usually multiple possible future trajectories given an input $x_i$ (although there is only one ground truth future trajectory $y_{i,gt}$ in a dataset recorded in the real world). To handle this situation, most state of the art methods (Salzmann et al., 2020) model a distribution of possible future trajectories $p_\theta(\mathbf{Y}_i \mid \mathbf{X}_i)$ to cover all the possibilities given the input $\mathbf{X}_i$ instead of predicting one trajectory. $\theta$ denotes the learning parameters of the model. To train such methods, most state-of-the-art models usually use maximum likelihood estimation (MLE) to maximize the likelihood of ground truth trajectory $\mathbf{Y}_{i,gt}$ in the predicted distribution. For example, the loss of CVAE-based model Trajectron++ (Salzmann et al., 2020) is Eq.1. This loss is used to maximize the lower bound of ground truth's likelihood when the coefficient $k = 1$.

$$
\begin{aligned}
L_{\text{traj++}} = & -\mathbb{E}_{\hat{\mathbf{z}} \sim q_{\theta 3}(\mathbf{z} \mid \mathbf{X}_i, \mathbf{Y}_{i,gt})}[\log p_{\theta 2}(\mathbf{Y}_{i,gt} \mid \mathbf{X}_i, \hat{\mathbf{z}})] \\
& + k D_{\text{KL}}(q_{\theta 3}(\mathbf{z} \mid \mathbf{X}_i, \mathbf{Y}_{i,gt}) \| p_{\theta 1}(\mathbf{z} \mid \mathbf{X}_i)) - I_q(\mathbf{X}_i; \mathbf{z}) \\
& \geq -\log p(\mathbf{Y}_{i,gt} \mid \mathbf{X}_i, \mathbf{Y}_{i,gt}) - I_q(\mathbf{X}_i; \mathbf{z}), \qquad \text{when} \ \ k = 1
\end{aligned}
\tag{1}
$$

**Limitation of MLE on Motion Forecasting**
MLE encourages the model to predict a distribution that allocates reasonable probability mass to the region where $\mathbf{Y_i}$ is located by minimizing the KL-divergence of predicted distribution and ground truth distribution. Because the domain of trajectory distribution is over the geometric locations, MLE makes these two distributions "close" to each other geometrically. However, we argue that maintaining the geometrical nearness only is not good enough for motion forecasting task in autonomous driving. In complex traffic scenarios, there can be many potential trajectories close enough to the ground truth geometrically but are very unlikely to happen. For example, if the ground truth trajectory $Y_{i,gt}$ is on the outermost lane, a trajectory that is close to $Y_{i,gt}$ but outside the drivable region is unlike to happen in the real world. However, MLE loss will not impose a significant enough penalty on such a case to avoid such a prediction. Fig.1 demonstrates a prediction example from Trajectron++ (Salzmann et al., 2020) where part of the distribution is outside of the derivable region or on the lane with the wrong direction. The MLE-based loss

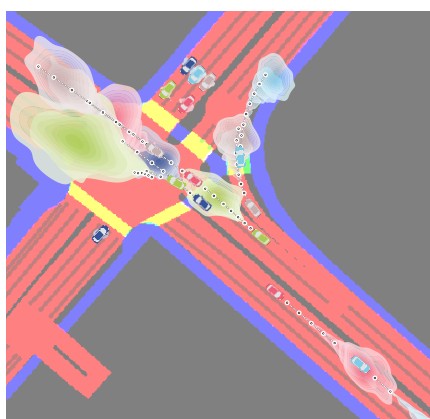

Figure 1: Examples of predicted distribution from Trajectron++ (Salzmann et al., 2020). White points denote the ground truth trajectory $\mathbf{Y}_{i,gt}$ and the color region indicates the predicted distribution. Some of the prediction go outside of drivable region or go to the lane in opposite direction.

only offers learning signals that contain the geometric location information of the ground truth trajectories. All the other contextual information, like the drivable region and the lane direction, are missing in the learning signals. While the inputs to a model contain rich contextual information, the model cannot use it to avoid the prediction that is geometrically close to ground truth but violates context. In contrast, this is quite a simple task for humans.

### 3.2 UNLIKELIHOOD LOSS

To mitigate this problem, we design a new loss term that encourages the model to consider the contextual information. Inspired by contrastive learning and unlikelihood training, we additionally

train our model to minimize the likelihood of trajectories that violate the contextual information given input $\boldsymbol{X}_i$. We denote them as negative trajectories $\boldsymbol{Y}_{i,neg}$. Let's first assume that we already have a distribution of negative trajectories $p_{\text{neg}}(\boldsymbol{Y}_i \mid \boldsymbol{X}_i)$. One intuitive way is to directly minimize the log likelihood of $\boldsymbol{Y}_{i,neg}$ in our predicted distribution, similar to MLE but in an opposite manner

$$L_{\text{unlike}} = \mathbb{E}_{\boldsymbol{X}_i, \sim \mathbb{D}, \boldsymbol{Y}_{i,neg} \sim p_{\text{neg}}(\boldsymbol{Y}_i | \boldsymbol{X}_i)}[\log p_\theta(\boldsymbol{Y}_{i,neg} \mid \boldsymbol{X}_i)] \tag{2}$$

However, the gradient of log function tends to infinity when the input tends to 0, which leads to unstable training since the model are optimized to minimize $p_\theta(\boldsymbol{Y}_{i,neg} \mid \boldsymbol{X}_i)$ and $p_\theta(\boldsymbol{Y}_{i,neg} \mid \boldsymbol{X}_i) \geq 0$. To avoid the infinity gradient region of log function, we add a small constant $\epsilon$ to the likelihood. The final loss term we propose is

$$L_{\text{unlike}} = \mathbb{E}_{\boldsymbol{X}_i, \sim \mathbb{D}, \boldsymbol{Y}_{i,neg} \sim p_{\text{neg}}(\boldsymbol{Y}_i | \boldsymbol{X})}[\log(p_\theta(\boldsymbol{Y}_{i,neg} \mid \boldsymbol{X}_i) + \epsilon))] \tag{3}$$

We call it unlikelihood loss. We use a coefficient $\gamma$ to balance $L_{\text{unlike}}$. The final training objective in case we combine our method with Trajectron++ is

$$L = L_{\text{traj++}} + \gamma L_{\text{unlike}} \tag{4}$$

Eq.4 is also easily adapted to combine with any other models that predict trajectory distribution as output. With the help of $L_{\text{unlike}}$, we inject the contextual information into the learning signal, force the model to better extract and use contextual information in $\mathbf{X}_i$, and generate more reasonable predicted distribution to avoid high $L_{\text{unlike}}$.

## 3.3 NEGATIVE TRAJECTORIES

Our proposed loss term is highly dependent on the negative samples $\boldsymbol{Y}_{i,neg}$ from the distribution $p_{\text{neg}}(\boldsymbol{Y}_i \mid \boldsymbol{X}_i)$. However, these are not given in the dataset. To solve this issue, we approximate the samples by directly drawing a set of trajectories from the predicted distribution and select the trajectories that violate the contextual information out by a context checker. Note that this checker does not need to be differentiable and it can be as complex and advanced as necessary. The type of unlike predictions the model learns to avoid by our method depends on the type of unlike trajectories the checker can detect.

**Design of Our Checker** We implement a map-based checker to judge whether a given trajectory suits the context or not. In detail, the checker examines whether the trajectory goes into the lane in the opposite direction or out of the road. We create a map that stores the lane direction at every location of lanes and the drivable region. Two examples are shown in Fig.2. We first check whether all the locations of a given trajectory are in the drivable region. If so, we further calculate angles between velocity and the lane direction at each time step to see whether they are all inside a 90 degree. The velocities are approximated by differentiating the trajectory. The trajectories that fail to pass the exam are judged as negative trajectories $\boldsymbol{Y}_{i,neg}$. Note that the lane direction information originally comes from the dataset and is usually incomplete or invalid like in the intersection. In this case, we only use the drivable region information. In addition, there are also a small part of ground truth trajectories in the dataset that violates the lane direction or drivable region. In this case, our checker skips the checking and we train it without $L_{\text{unlike}}$ to allow similar prediction. Note that this checker is not perfect due to the incomplete information and simple checking mechanism. A trajectory that passes the exam of the checker doesn't mean it is 100% compatible with the context. However, our method can still work properly, because our method only depends on the negative trajectories that do not pass the exam and has nothing to do with the passing ones. Our checker design offers reasonable negative trajectories to support our approach. But of course, a more advanced checker helps the model to avoid more complex unlikely prediction. We leave it open for future research.

## 3.4 ALGORITHM

The finial algorithm is shown in Alg.1. At each iteration, we first run the forward pass of the model to get the output distribution $p_\theta(\boldsymbol{Y}_i \mid \boldsymbol{X}_i)$ given the input $\boldsymbol{X}_i$. Then, $K$ negative candidate

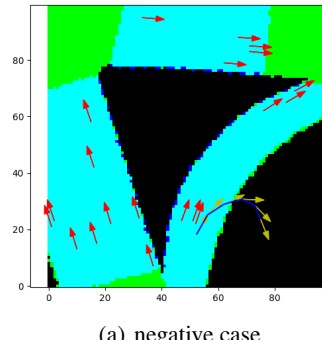 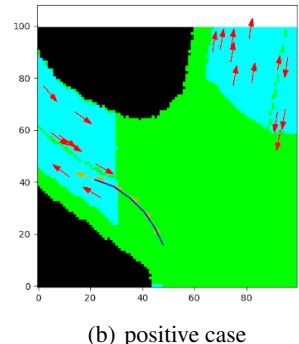

(a) negative case    (b) positive case

Figure 2: Examples of maps used in the checker in nuScences (Caesar et al., 2019a). Green and blue region together denote the drivable region and the blue means that we have lane direction information here. Random locations are sampled and their lane directions are plotted as red arrows to show the concrete directions. Blue line denotes the trajectories to check and the velocity directions are represented as yellow arrows. (a) shows a negative trajectory that goes out of the road. (b) is a passing case.

trajectories are drawn from this distribution and we select the negative trajectories out $\boldsymbol{Y}_{i,neg}$ via our checker. After that, ground truth trajectory and negative trajectories are used to calculate the loss function, update the model, and go to next iteration. Note that if there are no $\boldsymbol{Y}_{i,neg}$ in the $K$ negative candidates judged by our checker for data $i$, we don't apply $L_{\text{unlike}}$ on $i$.

## 3.5 GRADIENT ANALYSIS

Let's assume a single-mode prediction case that the future position $\boldsymbol{y}_{i,gt,t}$ at time step $t$ of agent $i$ is modeled by a simple Gaussian distribution $\mathcal{N}(\boldsymbol{y}_{i,t};\hat{\boldsymbol{\mu}}_{i,t},\hat{\sigma}_{i,t}\boldsymbol{I})$. $\hat{\boldsymbol{\mu}}$ and $\hat{\sigma}_{i,t}$ are calculated by the model. With a single negative position $\boldsymbol{y}_{i,neg,t}$ we define a simple loss for step $t$ $L_t = -\log \mathcal{N}(\boldsymbol{y}_{gt,t};\hat{\boldsymbol{\mu}}_t,\hat{\sigma}_t\boldsymbol{I}) + \log \mathcal{N}(\boldsymbol{y}_{neg,t};\hat{\boldsymbol{\mu}}_t,\hat{\sigma}_t\boldsymbol{I})$ and omit the subscript $i$ for brevity. The gradient of $L_t$ with respect to $\hat{\boldsymbol{\mu}}_t$ and $\hat{\sigma}_t$ in this case is (Derivation in Appx.A):

$$\frac{\partial L_t}{\partial \hat{\boldsymbol{\mu}}_t} = -\frac{1}{\hat{\sigma}_t^2}((\boldsymbol{y}_{gt,t} - \hat{\boldsymbol{\mu}}_t) + (\hat{\boldsymbol{\mu}}_t - \boldsymbol{y}_{neg,t})) \tag{5}$$

$$\frac{\partial L}{\partial \hat{\sigma}_t} = -\frac{1}{\hat{\sigma}_t^3}(||\boldsymbol{y}_{gt,t} - \hat{\boldsymbol{\mu}}_t||^2 - ||\boldsymbol{y}_{neg,t} - \hat{\boldsymbol{\mu}}_t||^2) \tag{6}$$

Eq.5 shows that the center of the predicted distribution $\hat{\boldsymbol{\mu}}_t$ is pushed towards $\boldsymbol{y}_{gt,t}$ and pushed away from $\boldsymbol{y}_{neg,t}$ by this learning objective. In Eq.6, when $\boldsymbol{y}_{gt,t}$ is closer to the center than $\boldsymbol{y}_{neg,t}$, $\frac{\partial L}{\partial \hat{\sigma}_t}$ is positive and $\hat{\sigma}_t$ is decreased. Note that $\boldsymbol{y}_{neg,t}$ is selected out from samples of $\mathcal{N}(\hat{\boldsymbol{\mu}}_t,\hat{\sigma}_t\boldsymbol{I})$, this means when $\mathcal{N}(\hat{\boldsymbol{\mu}}_t,\hat{\sigma}_t\boldsymbol{I})$ covers context-violated region and this region is farther than ground truth region, $\mathcal{N}(\hat{\boldsymbol{\mu}}_t,\hat{\sigma}_t\boldsymbol{I})$ will shrink to exclude the negative region and become a better estimation to the true data distribution.

When the prediction is not so accurate (e.g. at the beginning of training), our ground truth $\boldsymbol{y}_{gt,t}$ may be farther than the negative location $\boldsymbol{y}_{neg,t}$. In this case, $\mathcal{N}(\hat{\boldsymbol{\mu}}_t,\hat{\sigma}_t\boldsymbol{I})$ will expand to better cover the ground truth and make the prediction more uncertain. A simple approach to alleviate this issue is turning off our unlikelihood loss $L_{\text{unlike}}$ in the first few training epochs. We implement this by making $\gamma$ in Eq.4 as a sigmoid function centered at a specified epoch. By this way, we smoothly turn on $L_{\text{unlike}}$ during training.

## 4 EXPERIMENTAL RESULTS

In this section, we present the experimental results of our method to demonstrate our performance. Our method is easily to applied on state of the art models that generate a future trajectory distribution and can further improve their performance. In our experiments, we select Trajectron++ (Salzmann et al., 2020), one of the state-of-the-art methods on NuScenes dataset (Caesar et al., 2019b) with open-source implementation, as our base model. We extend the implementation to work on Argoverse dataset (Chang et al., 2019). We evaluate our approach on these two motion forecasting datasets.

**Algorithm 1:** Training process (Use Trajectron++ as base model)

---

Initialize the model parameters $\theta$;
Initialize learning rate $\alpha$ and coeifficient $\gamma$;
  **while** *not converge* **do**
    | $\mathbf{X}_i, \mathbf{Y}_{i,gt} \sim \mathbb{D}$
    | run forward pass to compute $p_\theta(\mathbf{Y}_i \mid \mathbf{X}_i)$
    | draw K trajectotries $\mathbf{Y}_{i,k} \sim p_\theta(\mathbf{Y}_i \mid \mathbf{X}_i)$
    | select $\mathbf{Y}_{i,neg}$ via checker
    | Compute $L_{\text{traj++}}$ using Eq.1
    | Compute $L_{\text{unlike}}$ using Eq.3
    | $L = L_{\text{traj++}} + \gamma L_{\text{unlike}}$
    | $\theta = \theta - \alpha \nabla_\theta(L)$
  **end**

---

**Test Model** Trajectron++ (Salzmann et al., 2020) is a CVAE-based (Sohn et al., 2015) model. Its input $\mathbf{X}_i$ contains positions, velocities, heading of the predicted and surrounding vehicles, and a map patch. The output distribution $p_\theta(\mathbf{Y}_i \mid \mathbf{X}_i) = \sum_{\boldsymbol{z}} p_{\theta 1}(\boldsymbol{z} \mid \mathbf{X}_i) p_{\theta 2}(\mathbf{Y}_i \mid \mathbf{X}_i, \boldsymbol{z})$ is a Gaussian mixture model with 25 components and modeled by an encoder net $p_{\theta 1}(\boldsymbol{z} \mid \mathbf{X}_i)$ and a decoder net $p_{\theta 1}(\mathbf{Y}_i \mid \mathbf{X}_i, \boldsymbol{z})$. In addition, it has another encoder net $q_{\theta 3}(\boldsymbol{z} \mid \mathbf{X}_i, \mathbf{Y}_{i,gt})$ used only in training. The original learning objective is shown in Eq.1. $I_q$ denotes the mutual information.

**Evaluation Metrics** we use average $l_2$ displacement error (ADE) and final $l_2$ displacement error (FDE) to evaluate the prediction performance. Each of them contains some sub-versions. FDE-1 is the FDE calculated using only 1 predicted trajectory. In both original Trajectron++ and our approach, this single trajectory is drawn from predicted distribution by greedy search step by step. ADE-Full/FDE-Full represents the quality of the whole output distribution. To compute ADE-Full/FDE-Full, we randomly sample 200 trajectories and calculate the average performance as the reported scores. In addition, we use our context checker to measure the context-violation rate in these 200 trajectories as a metric to show the context-related performance.

## 4.1 NUSCENES DATASET

nuScenes dataset (Caesar et al., 2019b) contains 1000 city driving scenes from both left-hand (Singapore) and right-hand (Boston) traffic regions. Each scene is 20s long and recorded in 2Hz. It is one of the biggest open-source motion forecasting datasets with detailed semantic maps.

**Experiments** The batch size is set to 1024. Models are trained for 35 epochs and we test the weights from the best epoch measured by average ADE on validation set. The coefficient $\gamma$ in Alg.1 increases gradually from 0 to 1 as a sigmoid function centered at 24th epoch. Initial learning rate is 3e-3 and it decays exponentially by 0.9995 per iteration. These hyperparameters except $\gamma$ are optimized for Trajectron++ and lead to better performance than that in original paper. In addition, we rotate the scenes randomly from $15°$ to $345°$ in the training set for data augmentation following the setting of original Trajectron++. For each model, we run 5 experiments and report the mean and standard deviation of the measured metrics. The models are trained to predict 3 seconds into the future. To evaluate on generalization beyond the training horizon, we test on both 3 second and 4 second prediction horizons.

Tab.1 shows the quantitative results 3s and 4s prediction measured by the FDE/ADE-Full metrics and the context-violation rate. Our unlikelihood loss improves Trajectron++ by about 8% for both metrics in both 3 and 4 second prediction horizons. Results indicate that our method helps improve the accuracy of the predicted distribution. This is also demonstrated in the qualitative comparison in Fig.3. The predicted distribution from our methods covers the not-drivable region less compared to original Trajectron++ without our proposed loss. In contrast, original Trajectron++ tends to violate the contextual information when prediction horizon is long. This shows that our method encourages the model to be more sensitive to the road boundary and the lane direction. In addition, we observe a reduction of the performance variance measured by standard deviation in Tab1, indicates that our

Table 1: **nuScenes:** Experiment results on nuScenes dataset (Caesar et al., 2019b) with Trajectron++ (Salzmann et al., 2020). FDE and ADE are averaged over 200 trajectories drawn from the predicted distribution. Our proposed loss improves the performance of Trajectron++ by about 8%, and avoid 16% context-violated prediction compared to Trajectron++, which indicates a better predicted distribution. Mean and standard deviation are calculated over 5 runs.

| Model | FDE-Full | | ADE-Full | | Context-Violation-Rate | |
|---|---|---|---|---|---|---|
| | 3s | 4s | 3s | 4s | 3s | 4s |
| **Trajectron++** | 1.46±0.07 | 2.74±0.10 | 0.59±0.04 | 1.04±0.05 | 7.29%±0.22% | 10.59%±0.54% |
| **Ours** | **1.34±0.04** | **2.51±0.06** | **0.54±0.02** | **0.95±0.03** | **6.57%±0.15%** | **8.85% ± 0.32%** |

Table 2: **nuScenes:** Experimental results on the nuScenes dataset for single prediction. FDE and ADE are computed by only one predicted trajectory. For both Trajectron++ and our method, this trajectory is sampled by greedy search. Our method helps to improve the predicted accuracy. Mean and standard deviation are calculated over 5 runs.

| Model | FDE-1 | | | |
|---|---|---|---|---|
| | 1s | 2s | 3s | 4s |
| Const. Velocity (Salzmann et al., 2020) | 0.32 | 0.89 | 1.70 | 2.73 |
| S-LSTM (Alahi et al., 2016) | 0.47 | - | 1.61 | - |
| CSP (Deo & Trivedi, 2018) | 0.46 | 2.35 | 1.50 | - |
| CAR-Net (Sadeghian et al., 2018) | 0.38 | - | 1.35 | - |
| SpAGNN (Casas et al., 2019) | 0.36 | - | 1.23 | - |
| Trajectron++ (Salzmann et al., 2020) | 0.07 | 0.45 | 1.14 | 2.20 |
| **Trajectron++ (our hyperparameters)** | 0.06±0.01 | 0.43±0.01 | 1.08±0.04 | 2.05±0.08 |
| **Ours** | **0.05±0.00** | **0.42±0.01** | **1.05±0.02** | **1.99±0.05** |

method helps to stable the training process. Comparison with other methods is shown in Tab.2. The FDE for single predicted trajectory is also improved by our method.

## 4.2 ARGOVERSE DATASET

Argoverse dataset (Chang et al., 2019) contains 300,000 5-second tracked scenarios in 2 American cities Miami and Pittsburgh. The data is recorded in 10 Hz. The first 2 seconds are used as input to predict the next 3 seconds future. It is also one of the biggest open-source motion forecasting datasets that offer semantic maps.

**Experiments** We downsample the data from 10Hz to 2Hz to make the setting similar to nuScenes following the setting of (Park et al., 2020). Argoverse does not release the ground truth future trajectories for the test dataset. Therefore, we use the original validation set as our test set in this experiments and randomly split the original training set into our training set with 95% data and our validation set with 5% data. Batch size is 256. Initial learning rate is 3e-3 and it decays exponentially by 0.9995 per iteration. The model is trained for maximal 60 epochs and we select the weights from the best epoch measured by average ADE in validation set. The coeffient $\gamma$ in Alg.1 increases gradually from 0 to 1 as a sigmoid function centered at 18th epoch. Trajectron++ is not designed for Argoverse. To make the experiments runnable, we made some small modifications that are explained in Appx.D. We execute 6 training instances for both original Trajectron++ and our method, report the average performance and the standard deviation. The results are listed in Tab.3. Compared to Trajectron++ without our method, our model improves the accuracy of the prediction by about 8% with our simple loss and reduces the performance variance by about 50% measured by standard deviation.

## 4.3 ABLATION STUDY

**Prediction horizon for negative candidates** Assume we have a predicted trajectory with a length six, and it is on the drivable region and obeys the lane direction. However, the trajectory tends to

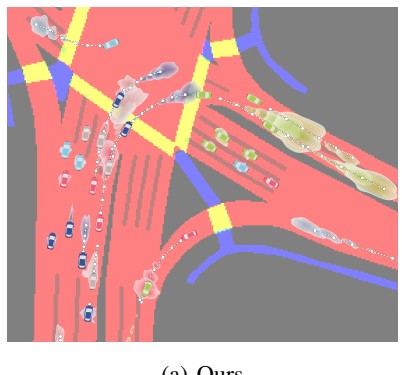 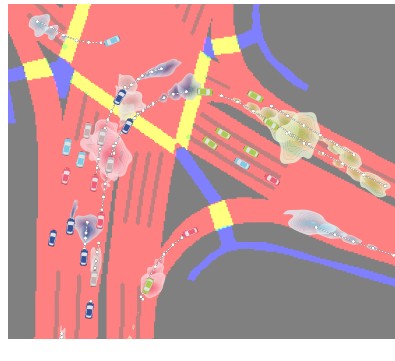

(a) Ours                                          (b) Trajectron++

Figure 3: Qualitative results of our method and Trajectron++ in a complex scenario. Some of the predicted distribution of Trajectron++ are out of the road or cover the lane with wrong direction. Our method helps alleviate this issue. Predicted distribution is plotted as colored region and white points denotes the ground truth trajectories. More results are in Appx.F

Table 3: **Argoverse:** Experimental results on Argoverse dataset. Compared to our base model Trajectron++, our proposed method helps increase the accuracy and stable the performance.

| Model | ADE-1 | FDE-1 | ADE-Full | FDE-Full |
|---|---|---|---|---|
| **Trajectron++** | 1.15±0.14 | 2.73±0.24 | 1.43±0.15 | 3.40±0.20 |
| **Ours** | **1.06±0.08** | **2.58±0.14** | **1.34±0.06** | **3.25±0.10** |

Table 4: **nuScenes:** Ablation study on prediction horizon of negative candidates on nuScenes dataset.

| $Model$ | $PH_n$ | FDE Full 1s | 2s | 3s | 4s | FDE ML 1s | 2s | 3s | 4s | B. Violations 1s | 2s | 3s | 4s |
|---|---|---|---|---|---|---|---|---|---|---|---|---|---|
| Trajectron++ | - | 0.107 | 0.560 | 1.378 | 2.621 | 0.052 | 0.396 | 1.010 | 1.950 | 9.169% | 9.710% | 13.048% | 21.369% |
| Ours | 3s | 0.105 | 0.538 | 1.320 | 2.494 | 0.054 | 0.414 | 1.029 | 1.960 | 9.188% | 9.619% | 11.818% | 17.460% |
| Ours | 4s | **0.087** | **0.498** | **1.238** | **2.339** | **0.050** | **0.387** | **0.981** | **1.869** | 9.183% | 9.620% | 11.872% | 17.551% |
| Ours | 5s | 0.087 | 0.497 | 1.259 | 2.406 | 0.056 | 0.397 | 1.002 | 1.889 | 9.174% | 9.667% | 12.106% | 17.992% |
| Ours | 6s | 0.099 | 0.521 | 1.297 | 2.474 | 0.059 | 0.405 | 1.004 | 1.912 | 9.170% | 9.636% | 12.046% | 17.761% |

hit the road boundary in the near future (e.g., in 1 second). Such a trajectory can pass our checker's exam, but it is still unlikely to happen in the real world. We can easily select out such a trajectory by extending the prediction horizon for the candidate trajectories and examining our checker's extended version. To verify whether this helps us build a better checker, we extend the prediction horizon for negative candidates from 3 seconds to 4, 5, 6 seconds, respectively, and examine them by our original checker. The selected negative trajectories are truncated back to 3 seconds for computing $L_{\text{unlike}}$. We can see in table 4 that the model benefits from an adequately extended prediction horizon. The prediction horizon for ground truth trajectory is 3s. By extending the negative trajectories 1 second more, we improve the prediction accuracy. Numbers are averaged over two training instances.

## 5 CONCLUSION

We present unlikelihood guided trajectory prediction method, that minimizes the probability of unlikely trajectories. During training, our context checker detects predicted unlikely trajectories and their probabilities are reduced through an unlikelihood loss. Our method can be incorporated into state-of-the-art models with a maximum likelihood estimation objective. Our experimental results demonstrate that our method significantly improves state-of-the-art trajectory prediction models.

We hope that our work may encourage future work on exploring better unlikelihood methods for trajectory prediction and improved context checker models.

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

## A    DERIVATION OF GRADIENT

Here we show how to obtain Eq.5 and Eq.6

$$L_t = -\log \mathcal{N}(\boldsymbol{y}_{gt,t}; \hat{\boldsymbol{\mu}}_t, \hat{\sigma}_t \boldsymbol{I}) + \log \mathcal{N}(\boldsymbol{y}_{neg,t}; \hat{\boldsymbol{\mu}}_t, \hat{\sigma}_t \boldsymbol{I})$$
$$= \frac{1}{2}(\log 2\pi + \log \hat{\sigma}_t^2 + \frac{||\boldsymbol{y}_{gt,t} - \hat{\boldsymbol{\mu}}_t||^2}{\hat{\sigma}_t^2}) - \frac{1}{2}(\log 2\pi + \log \hat{\sigma}_t^2 + \frac{||\boldsymbol{y}_{neg,t} - \hat{\boldsymbol{\mu}}_t||^2}{\hat{\sigma}_t^2})$$
$$= \frac{1}{2\hat{\sigma}_t^2}(||\boldsymbol{y}_{gt,t} - \hat{\boldsymbol{\mu}}_t||^2 - ||\boldsymbol{y}_{neg,t} - \hat{\boldsymbol{\mu}}_t||^2)$$

$$\frac{\partial L_t}{\partial \hat{\boldsymbol{\mu}}_t} = \frac{\partial}{\partial \hat{\boldsymbol{\mu}}_t} \frac{1}{2\hat{\sigma}_t^2}(||\boldsymbol{y}_{gt,t} - \hat{\boldsymbol{\mu}}_t||^2 - ||\boldsymbol{y}_{neg,t} - \hat{\boldsymbol{\mu}}_t||^2)$$
$$= -\frac{1}{\hat{\sigma}_t^2}((\boldsymbol{y}_{gt,t} - \hat{\boldsymbol{\mu}}_t) + (\hat{\boldsymbol{\mu}}_t - \boldsymbol{y}_{neg,t}))$$

$$\frac{\partial L}{\partial \hat{\sigma}_t} = \frac{\partial}{\partial \hat{\sigma}_t} \frac{1}{2\hat{\sigma}_t^2}(||\boldsymbol{y}_{gt,t} - \hat{\boldsymbol{\mu}}_t||^2 - ||\boldsymbol{y}_{neg,t} - \hat{\boldsymbol{\mu}}_t||^2)$$
$$= -\frac{1}{\hat{\sigma}_t^3}(||\boldsymbol{y}_{gt,t} - \hat{\boldsymbol{\mu}}_t||^2 - ||\boldsymbol{y}_{neg,t} - \hat{\boldsymbol{\mu}}_t||^2)$$

## B    GRADIENT ANALYSIS IN GAUSSIAN MIXTURE MODEL

In case we model the output distribution as a Gaussian mixture model $p_{\text{GMM}}(\boldsymbol{y}_t) = \sum_i \phi_i \mathcal{N}(\boldsymbol{y}_t; \hat{\boldsymbol{\mu}}_{i,t}, \hat{\sigma}_{i,t} \boldsymbol{I})$, the gradient of $L_t$ w.r.t. the mean of component $i$ is

$$\frac{\partial L_t}{\partial \hat{\boldsymbol{\mu}}_{i,t}} = \frac{\partial}{\partial \hat{\boldsymbol{\mu}}_{i,t}}(-\log \frac{p_{\text{GMM}}(\boldsymbol{y}_{gt,t})}{p_{\text{GMM}}(\boldsymbol{y}_{neg,t})})$$
$$= -\phi_i(\frac{1}{p_{\text{GMM}}(\boldsymbol{y}_{gt,t})} \frac{\partial \mathcal{N}(\boldsymbol{y}_{gt,t}; \hat{\boldsymbol{\mu}}_{i,t}, \hat{\sigma}_{i,t} \boldsymbol{I})}{\partial \hat{\boldsymbol{\mu}}_{i,t}} - \frac{1}{p_{\text{GMM}}(\boldsymbol{y}_{neg,t})} \frac{\partial \mathcal{N}(\boldsymbol{y}_{neg,t}; \hat{\boldsymbol{\mu}}_{i,t}, \hat{\sigma}_{i,t} \boldsymbol{I})}{\partial \hat{\boldsymbol{\mu}}_{i,t}})$$
$$= -\frac{\phi_i}{\sigma_{i,t}^2}(\frac{\mathcal{N}(\boldsymbol{y}_{gt,t}; \hat{\boldsymbol{\mu}}_{i,t}, \hat{\sigma}_{i,t} \boldsymbol{I})}{p_{\text{GMM}}(\boldsymbol{y}_{gt,t})}(\boldsymbol{y}_{gt,t} - \hat{\boldsymbol{\mu}}_{i,t}) - \frac{\mathcal{N}(\boldsymbol{y}_{neg,t}; \hat{\boldsymbol{\mu}}_{i,t}, \hat{\sigma}_{i,t} \boldsymbol{I})}{p_{\text{GMM}}(\boldsymbol{y}_{neg,t})}(\boldsymbol{y}_{neg,t} - \hat{\boldsymbol{\mu}}_{i,t}))$$

This gradient shows that the center $\hat{\boldsymbol{\mu}}_{i,t}$ of component $i$ will be pushed towards the ground truth location $\boldsymbol{y}_{gt,t}$ and way from the negative location $\boldsymbol{y}_{neg,t}$. For $\hat{\sigma}_{i,t}$ we have

$$\frac{\partial L}{\partial \hat{\sigma}_{i,t}} = -\phi_i(\frac{1}{p_{\text{GMM}}(\boldsymbol{y}_{gt,t})} \frac{\partial \mathcal{N}(\boldsymbol{y}_{gt,t}; \hat{\boldsymbol{\mu}}_{i,t}, \hat{\sigma}_{i,t} \boldsymbol{I})}{\partial \hat{\sigma}_{i,t}} - \frac{1}{p_{\text{GMM}}(\boldsymbol{y}_{neg,t})} \frac{\partial \mathcal{N}(\boldsymbol{y}_{neg,t}; \hat{\boldsymbol{\mu}}_{i,t}, \hat{\sigma}_{i,t} \boldsymbol{I})}{\partial \hat{\sigma}_{i,t}})$$
$$= -\frac{\phi_i}{\sigma_{i,t}^3}(\frac{\mathcal{N}(\boldsymbol{y}_{gt,t}; \hat{\boldsymbol{\mu}}_{i,t}, \hat{\sigma}_{i,t} \boldsymbol{I})}{p_{\text{GMM}}(\boldsymbol{y}_{gt,t})}(||\boldsymbol{y}_{gt,t} - \hat{\boldsymbol{\mu}}_{i,t}||^2 - \sigma_{i,t}^2)$$
$$- \frac{\mathcal{N}(\boldsymbol{y}_{neg,t}; \hat{\boldsymbol{\mu}}_{i,t}, \hat{\sigma}_{i,t} \boldsymbol{I})}{p_{\text{GMM}}(\boldsymbol{y}_{neg,t})}(||\boldsymbol{y}_{neg,t} - \hat{\boldsymbol{\mu}}_{i,t}||^2 - \sigma_{i,t}^2))$$

## C    THE INFLUENCE OF OUR LOSS WITHOUT MAP INPUT

In this experiment, we remove the map input to the model and demonstrate the influence of our unlikelihood loss in this case. Results are shown in Tab.5 Interestingly, our loss helps improve the performance from 1.52 to 1.42 measured by the FDE-Full in 3 seconds case although the model doesn't see the context during inference. We think the reason is that compared to the Trajectron++

Table 5: The influence of our loss without map input to the model on nuScenes. Compared to Trajectron++ without map input, our loss performs better, which indicates that our loss can inject context information into learning signals

| Model | FDE-Full 3s | ADE-Full 3s |
|---|---|---|
| **Trajectron++ without map input** | 1.52±0.04 | 0.61±0.02 |
| **Trajectron++** | 1.46±0.07 | 0.59±0.04 |
| **Ours without map input** | 1.42±0.03 | 0.57±0.01 |
| **Ours** | 1.34±0.04 | 0.54±0.02 |

without map input, our unlikelihood loss still offers context information to support the training since this loss is calculated using the context. Therefore, our model receives more information during training and performs better. Besides, our method without map input even achieves a comparable result (FDE-FUll 1.42) compared to Trajectron++ with map input (FDE-Full 1.46). This experiment shows that our loss can inject context information into learning signals. In addition, map input improves FDE-Full of Trajectron++ in 3s prediction from 1.52 to 1.46, which is about 6 cm. However, when we further add our unlikelihood loss, performance improved from 1.46 to 1.34, which is 12 cm. Our loss triple the contribution of context information, which is significant.

## D    MODIFICATION ON TRAJECTRON++ FOR ARGOVERSE EVALUATION

The original Trajectron++ (Salzmann et al., 2020) is evaluated on nuScenes dataset (Caesar et al., 2019b) but not on Argoverse (Chang et al., 2019). To make it runnable on Argoverse, we make some modifications on Trajectron++. The input states of the full version of Trajectron++ consist of the positions, velocities, accelerations, heading angles and heading angular velocities. However, Argoverse doesn't offer the data for heading angles and heading angular velocities. Therefore, we remove them from the input states. In addition, Trajectron++ has a unicycle physics model to convert the predicted velocity from neural networks to locations. The unicycle physics model requires also the angular velocities information. Therefore, we replace the unicycle model by the single integrator model that requires only velocities and initial positions. The maps offered by nuScens and Argoverse are different, too. We stack the drivable region and region of interest offered by Argoverse and upsample them to the same resolution as nuScenes as the input map. Please refer to Argoverse for the details. Lastly, Trajectron++ use separate modules to process surrounding vehicles and surrounding pedestrians in nuScenes. However, Argoverse doesn't offer the labels for the surrouding agents and just simply denote them as "others". Therefore, we remove the pedestrian modules in Trajectron++ and use the vehicle modules to process both surrounding pedestrians and vehicles.

# E    COMPARISON WITH OTHER METHODS ON ARGOVERSE

Our experiment setting on Argoverse follows AttGlobal-CAM-Nf (Park et al., 2020). Here we list the detailed comparison with their numbers using their evaluation metrics minADE-12 and minFDE-12, which are the minimal ADE and FDE over 12 prediction candidates, in Tab.6. Both Trajectron++ and our method outperform their numbers. In addition, our unlikelihood loss helps improve the performance of Trajectron++. Our scores and the standard deviation are calculated over 4 training instances.

Table 6: Experimental results on Argoverse dataset

| Model | minADE-12 | minFDE-12 |
|---|---|---|
| CSP (Deo & Trivedi, 2018; Park et al., 2020) | 1.39 | 2.57 |
| DESIRE (Lee et al., 2017; Park et al., 2020) | 0.90 | 1.45 |
| MATF-GAN (Zhao et al., 2019; Park et al., 2020) | 1.26 | 2.31 |
| R2P2-MA (Rhinehart et al., 2019; Park et al., 2020) | 1.11 | 1.77 |
| AttGlobal-CAM-Nf (Park et al., 2020) | 0.73 | 1.12 |
| **Trajectron++** (Salzmann et al., 2020) | 0.65±0.09 | 1.08±0.07 |
| **Ours** | **0.63±0.04** | **1.08±0.06** |

# F    QUALITATIVE RESULTS

Here, we demonstrate our method's qualitative results compared with Trajectron++ for 3 seconds prediction. We randomly sample 50 trajectories from the predicted prediction, use kernel density estimation (KDE) to approximate the total output distribution from the samples, and print it out in Fig.4. White points represent the ground truth trajectories. Compared to Trajectron++, our method suits the contextual information more and therefore is more accurate and plausible.

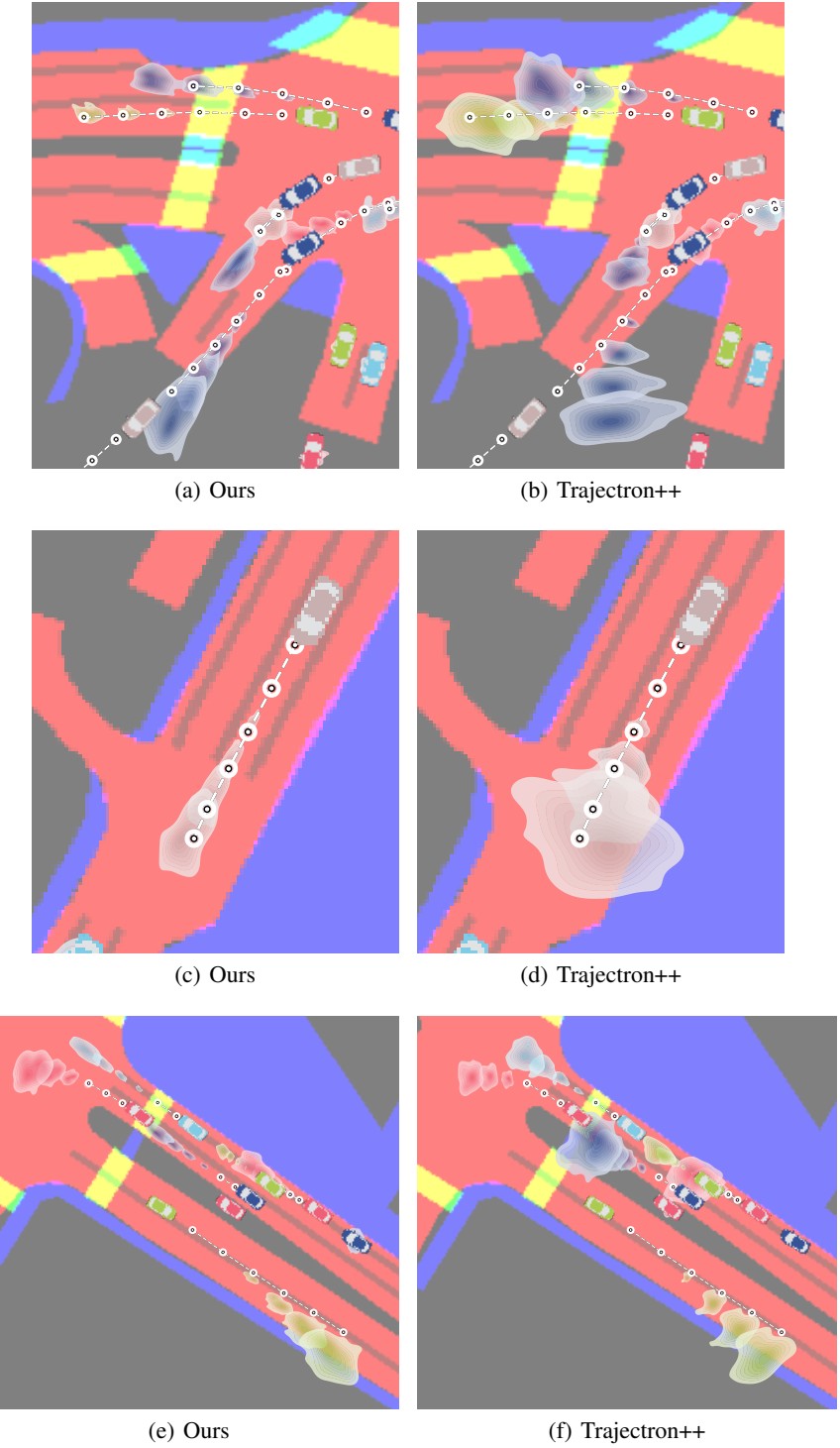

(a) Ours

(b) Trajectron++

(c) Ours

(d) Trajectron++

(e) Ours

(f) Trajectron++

Figure 4: Qualitative results of our method and Trajectron++.

