# OpenReview forum: "Motion Forecasting with Unlikelihood Training"
_ICLR.cc/2021/Conference — Reject_

### Official Review · AnonReviewer1 · 2020-10-26
**Insignificant results, insufficient evaluation, low novelty --- strong reject**

**Rating:** 4
**Confidence:** 5

**Review:**

I work extensively and publish regularly in autonomous vehicle trajectory prediction and am well-placed to provide a review for this paper.

*Summary:* The paper addresses the problem of trajectory prediction, which is to predict the future trajectory of a vehicle given its current trajectory, taking into account the nearby vehicle data. The main contribution of the paper is to investigate the possibility of improving performance of trajectory prediction using negative trajectories, that are generated heuristically.

This paper suffers from some major flaws in three key areas -- 1) Insignificance of results, 2) Insufficient Evaluation and 3) Low technical novelty. I will be talking about all of these separately below:

**Insignificance of results:** The main contribution of the paper is the uncertainty loss term. Therefore, in order to evaluate the contribution of this loss term, it would need to be evaluated against methods that use the same hyperparameters in a loss equation that excludes the uncertainty loss term. That would be the trajectron++ (using the current paper's hyperparameters) (Tab.2, second last row). The maximum average improvement is 6 cm. To put this in perspective, this improvement is less than a *quarter* of the width of a standard vehicle tire. The advantages and usefulness of improvements on such a small scale are therefore questionable.

This insignificance of results is not just contained in Tab.2. In Tables 1 and 3, the maximum average improvement is 9cm. This is likewise not useful. I have worked with several codebases that do trajectory prediction and I can say from experience that improvements on such small scales can easily be achieved simply by tweaking hyperparameters. Therefore, I have serious doubts whether the results are significant with respect to the proposed contribution or are just a consequence of model tuning.

*Conclusion*: The current results have not convinced me even slightly if the proposed uncertainty loss makes any difference to the current SOTA in trajectory prediction

**Insufficient Evaluation:**  Trajectory prediction using deep neural nets is a very well studied field with dozens of benchmark approaches now that report results using the ADE/FDE metrics. So any contribution would need to be evaluated against multiple methods. In this paper, the proposed contribution has only been evaluated against one method.

It should be clearly noted that to specifically measure the value of the contribution, other than trajectron++ ("our hyperparameters"), any of the other baselines are not relevant as the evaluation needs to be performed keeping all the parameters same, which is not the case with any of the other methods. (If the parameters are different, then it cannot be ascertained whether the gains come from the uncertainty loss term, or from somewhere else)

*Conclusion:* The validity of the method is questionable since it is compared to only one baseline, when there are more than a dozen SOTA methods available.

**Low Technical Novelty:** The novelty of the approach consists of the so-called "Context Checker" which identifies negative trajectories. However, the basis on which these negative trajectories are identified, as well as the definition of negative trajectories, is deeply heuristic and contrived.

To begin with, trajectory prediction consists of a distribution of future possibilities. The entire distribution contributes to the uncertainty. So the proposed approach wherein the authors manually select a only a part of this distribution, give it a fancy name like "negative trajectory", and re-train it without a formal explanation seems heuristic and unsound.

Another concern that I immediately registered was the number of such negative trajectories, which would be negligible compared to the entire distribution of future trajectories. In fact, this is even explicitly agreed upon by the authors themselves in Section 3.3 ("Design of our Checker"). The low number of such "negative" examples would do little to impact the performance, which corroborates my earlier point about the insignificance of results.

Finally, the entire approach is a simple and heuristic extension of trajectron++. The extension involves simply re-labeling a part of the predictions of trajectron++ and re-training them. This is not enough to meet the standards of ICLR.

*Conclusion:* The incremental extension over Trajectron++ and heurisitc and unsound nature of the novelty.
___
**Overall:** My overall recommendation for paper is a **strong reject** since according to my review, it fails on all of the crucial parameters used to judge a paper.

=========== Updated Score============

Of the three concerns above, the authors satisfactorily addressed the point related to technical novelty, hence I am increasing my score.

---

> ### Author Response · Authors · 2020-11-23
> **Answers to Questions Part 1**
>
> We thank Reviewer 1 for the feedback. We find the review does not give credit to several efforts in the paper based on which we think there could be a misunderstanding.
>
> ***In Tab.2, the maximum average improvement is 6 cm. In Tables 1 and 3, the maximum average improvement is 9cm. This is likewise not useful. The advantages and usefulness of improvements on such a small scale are therefore questionable.***
>
> The overall improvement is about 3-5 \% when our method is applied. Our method focuses more on better utilizing the context information to improve the prediction quality by making the prediction follows context more. As metrics ADE/FDE are based on the geometric distance only, they cannot directly reflect the performance in the context. Therefore, we additionally evaluate our method by the context violation rate measured by our context checker and updated our experiment section to include these results (see Table 1 in the revised paper).  Our method helps reduce the violation rate from 10.6\% to 8.9\%. About 16\% context-violated prediction is avoided by our method.
>
>
> ***I have serious doubts whether the results are significant with respect to the proposed contribution or are just a consequence of model tuning.***
>
> The hyperparameters that both our method and Trajectron++ use are tuned w.r.t. Trajectron++ and are the same to both our method and Trajectron++. Therefore, we don't think our improvement is a consequence of model tuning. Besides, Tab. 2 contains the performance of both the original number of Trajectron++ and the number after model tuning, the consequence of model tuning is already shown here. Finally, the small standard deviation of our performance shows that our improvement is solid.
>
>
> ***Trajectory prediction consists of a distribution of future possibilities. The entire distribution contributes to the uncertainty. So the proposed approach wherein the authors manually select only a part of this distribution, give it a fancy name like "negative trajectory", and re-train it without a formal explanation seems heuristic and unsound.***
>
> Similar techniques are used in NLP domain (Welleck et al., 2019).
> Our idea is to reduce the probability density of the region that violates contextual information in the predicted distribution, so as to encourage the model respects contextual information more.  We do it in the opposite direction of MLE, where MLE increases the likelihood of ground truth trajectory, and our method additionally decreases the likelihood of trajectories that violate contextual information. We don't re-train the model. Our unlikelihood loss and the normal MLE loss are applied at the same time. I think our loss is intuitive to reduce the probability density of the region that violates contextual information in the predicted distribution.
> In addition, we provide a mathematical gradient analysis to show how our method affects the prediction in case the prediction is a Gaussian distribution in section 3.5. In conclusion, we don't think our method is unsound.
>
> ***The number of such negative trajectories would be negligible compared to the entire distribution of future trajectories. In fact, this is even explicitly agreed upon by the authors themselves in Section 3.3 ("Design of our Checker"). The low number of such "negative" examples would do little to impact the performance, which corroborates my earlier point about the insignificance of results.***
>
> According to the reduced context violation rate, our method is not negligible.
> If we think about the number of ''positive'' trajectories people use for training in previous works, it is 1 since we only have 1 ground truth trajectory.
> But the training still works by increasing the likelihood of ground truth trajectory in the predicted distribution. In most cases, we have more than 1 negative trajectory. Our method decreases the likelihood of negative trajectories in the distribution. We think this is not a problem.
>
> Indeed, our method does nothing in the training cases where we don't have any negative trajectories. But this is expected since
> in these cases the predicted distribution is already good in the perspective of following contextual information. Therefore, our method doesn't need to affect this predicted distribution further. Our method focuses more on cases where predicted distribution violates the contextual information.

---

> ### Author Response · Authors · 2020-11-23
> **Answers to Questions Part 2**
>
> ***The basis on which these negative trajectories are identified, as well as the definition of negative trajectories, is deeply heuristic and contrived.
> The entire approach is a simple and heuristic extension of trajectron++. The extension involves simply re-labeling a part of the predictions of trajectron++ and re-training them. This is not enough to meet the standards of ICLR.***
>
> Could you please clarify why our method is contrived?
> Most previous works focus on how to better predict the correct trajectories. However, the problem of how to avoid bad predictions is less explored before.
> We believe this is an important problem and we target this problem by recognizing the poorly-performed outputs and penalize them. The technical details of our approach are simple.
> However, our method is an effective and principled way to better utilize contextual information to avoid bad predictions and therefore improves the model's quality.

---

### Official Review · AnonReviewer4 · 2020-10-28
**Hard negative mining for improved trajectory forecasting**

**Rating:** 5
**Confidence:** 4

**Review:**

The paper proposes an approach to improving the accuracy of trajectory prediction by adding a loss that minimizes the likelihood of trajectories that violate the contextual constraints. The negative trajectories themselves are sampled from the current version of the prediction model and are limited to those that violate the contextual constraints, such as the lane direction and driving area.

Authors show that their method improves state-of-the-art Trajectron++ on the widely used NuScenes dataset. Additionally, they compare to their baseline method on the Agroverse dataset, but due to absence of other results on the Agroverse dataset, this serves more as an ablation study.

My biggest concern is the absence of larger evaluation on popular benchmarks such as ETH and UCY. Many of the methods that authors compare with, including Trajectron++ and Social LSTM, provide results on those benchmarks and evaluation on those could strengthen the paper.

My preliminary evaluation is 5 because while authors show improvements for one particular method:
- The evaluation on ETH / UCY is lacking
- The evaluation showing improvements when combining with other trajectory prediction methods is lacking
- Novelty is limited since the idea itself has already been applied a number of times, both referenced by the authors for the NLP domain (+[Sequence level training with recurrent neural networks, ICLR'16]) and in the object tracking / trajectory prediction domain ([Eliminating Exposure Bias and Metric Mismatch in Multiple Object Tracking, CVPR'19])

Typos:
abstract: (that conflicts --> that conflict)
page 1: (checker is used to cull out --> to cuT out)
page 4, sec. 3.3 (there are not given -> theSe are not given)
page 4, sec. 3,3 (In detail, The checker --> In detail, the checker)

---

> ### Author Response · Authors · 2020-11-22
> **Answers to Questions**
>
> We thank Reviewer 4 for the valuable feedback reviews. We address here the questions.
>
> ***My biggest concern is the absence of larger evaluation on popular benchmarks such as ETH and UCY. ***
>
> This paper is mainly about how to better use context information to improve the predicted distribution in the field of autonomous driving. The 2 pedestrian-only benchmarks ETH and UCY contain almost no context information: UCY only provides the trajectory data without any context. ETH offers super simple maps (only contains 1 or 2 walls). They are not suitable for the task we focus on.
>
>
> ***Novelty is limited since the idea itself has already been applied a number of times, both referenced by the authors for the NLP domain (+[Sequence level training with recurrent neural networks, ICLR'16]) and in the object tracking/trajectory prediction domain ([Eliminating Exposure Bias and Metric Mismatch in Multiple Object Tracking, CVPR'19])***
>
> We focus on the problem of how to let the model better utilize the contextual information to avoid context-violated prediction.
> We believe, our work is the first method that encourages the model to respect the contextual information by unlikelihood training.
> In addition, most previous works focus on how to better predict the correct trajectories. However, the problem of how to avoid bad prediction is less explored before. We think we provide a simple but principle method to alleviate this problem.
>
>
> ***Typo***
>
> Thanks for your pointer! We have revised our paper to fix it.

---

### Official Review · AnonReviewer3 · 2020-10-28
**Review [Updated]**

**Rating:** 4
**Confidence:** 5

**Review:**

**SUMMARY**

The present paper considers the problem of context integration in probabilistic agent trajectory predictors, particularly Trajectron++. It starts with the observation that these predictors often do a bad job at considering non-drivable areas in their predictions even if context information is injected as part of the input. The paper proposes adjusting the loss function by adding an unlikelihood loss term. This term modifies existing MLE-type losses in a way that discourages context violations (in the present case, it discourages prediction of trajectories that are on non-drivable space).
The unlikelihood loss is integrated into Trajectron++ and it is demonstrated that the resulting variant of Trajectron++ not only improves upon the prediction error but also reduces the likelihood mass in regions violating the context cues.

**STRENGTHS**
- The paper addresses a very important problem for probabilistic trajectory prediction. Predictors regularly violating context cues may lead to very dangerous downstream behavior in a driving stack. This makes the general problem of investigating different means of context integration highly relevant.
- The general idea of incorporating context via a combination of a checker and a modified loss term is cool and seems to provide a very strong prior.

**WEAKNESSES**
- The evaluation does not sufficiently support the author's claim that the proposed method is generally useful (and to what extent) beyond Trajectron++.
- With the field moving fast, most of the baseline models are not state-of-the-art anymore. This is not necessarily a major problem as the idea has value for itself. It's just that the current formulation of the abstract might suggest that the proposed work is the best performing predictor. I am not sure if this is still true.
- The unlikelihood loss term itself and adding the $\varepsilon$ term within the log seem both to have no theoretical justification (if there is one, please provide a reference. I looked only briefly but Welleck et al.'s original work seem not to have one either). It is important to keep in mind that changing the loss also changes the underlying distribution assumption. In the present case, this has potential implications for how consistent the network's prediction of the variance is.

**CLARITY**

Overall, the paper is clearly written. I have only some smaller questions/remarks:
- given that several new prediction papers are coming out each month, it seems to make almost no sense to have a "trajectory prediction" section in related work. Maybe refer to a recent survey and then discuss a subset of methods that incorporate context in different ways. E.g. top-down images, vectors, latent biases, ...
- The authors write "When the prediction is not so accurate (e.g. at the beginning of training), our ground truth $y_{gt,t}$ may be farther than the negative location $y_{neg,t}$. In this case, $N(\hat{\mu}_t; \hat{\sigma}_t$) will expand to better cover the ground truth and make the prediction more uncertain". I am not sure I fully understand why this is happening. Could the authors elaborate on that?
- When describing FDE-1, the authors write "In both original Trajectron++ and
our approach, this single trajectory is drawn from predicted distribution by greedy search step by step". What does the search look for? The closest sampled trajectory? If yes, why not simply use the mean?

**REPRODUCIBILITY**

I believe this work to be reproducible. I also applaud the authors for planning to make their code available.

**CORRECTNESS**
- The claim "a distribution that allocates reasonable probability mass to the region where Yi is located by minimizing the KL-divergence of predicted distribution and ground truth distribution." is in its generality not true. MLE does not require the use of KL-Divergences. This only happens in some models, particularly those that use (ammortized) variational inference such as VAEs.
- In eq. 3, simply adding the small constant seems like a hack. I usually don't mind those but in the present case, the hack has a strong impact on what it is that the model is ultimately optimizing.
- I am not sure if the final loss (4) proposed in this work gives rise to any meaningful probabilistic interpretation.
- The authors claim that "Eq.4 is also easily adapted to combine with any other models".  This is not necessarily true. While the proposed loss can be integrated in most other models, the adaptation is not necessarily easy and obvious. For example, how would one adapt it to a GAN based predictor? I can think of several options (e.g. as a loss on the generator, by providing additional information to the discriminator, ...) neither of which is the obvious right choice. Also, the work assumes the availability of an output distribution to draw K negative candidate trajectories from. However, some predictors merely output a single trajectory (or a predefined small number of trajectories) and not an entire distribution. They may not contain K negative examples.
- The new loss term might induce an undesired bias for scenarios where violating context would be the right thing to do and the reasonable thing to predict. Thus, it would be interesting to see how the approach performs in such cases in comparison to the baseline.

**EVALUATIONS**
- The claim that the "method can be viewed as a simple add-on to any models that estimate the distribution of future trajectories." should be justified by experiments on multiple predictors and not only Trajectron++. In general, I would recommend focusing evaluations more on comparing different methods of context integration in trajectory prediction.
-  It is also claimed that the "coefficient gamma in Alg.1 increases gradually from 0 to 1 as a sigmoid function centered at 24th epoch." Is this rescaled in some way to ensure the graduality?
- "To evaluate on generalization beyond the training horizon, we test on both 3 second and 4 second prediction horizons." This testing protocol seems interesting to me but I wonder if it is meaningful. I would argue that in practice the prediction horizon can be known at training time and predictors do not need to generalize to other horizons but I am curious about the authors' thoughts on this.
- In the same spirit, shouldn't the split be between different locations? I.e., are all vehicles in NuScene's test set from different locations as in the training set? Same question for the split you used in Argoverse? I couldn't find this information on their websites. For the present work, this seems important to judge whether the network learned to actually deal with different context or if unlikelihood training supported overfitting to existing locations.
- Could you mention how many iterations there are per epoch. This would simplify understanding the extent of learning rate decay.
- How was $\epsilon$ from eq (3) chosen? Did different choices have an effect on the training outcome?
- It would be interesting to see the tradeoff between using this type of training and injecting a top-down image. Would it be possible to train Trajectron in a way that does not involve context-input as part of the ablation studies?
- In App B. the authors write that "The input states of the full version of Trajectron++ consist of the positions, velocities, accelerations, heading angles and heading angular velocities. However, Argoverse doesn’t offer the data for heading angles and heading angular velocities. Therefore, we remove them from the input states. [...] we replace the unicycle model by the single integrator model that requires only velocities and initial positions" I am not sure if this is the fair thing to do. If understood correctly, NuScenes does not contain some of this information either. Instead, it is computed via some of NuScenes' devkit helper function (see e.g. [here](https://github.com/nutonomy/nuscenes-devkit/blob/d8403d35a49f9a5f2b8707129c8af1eff6a8906c/python-sdk/nuscenes/prediction/helper.py#L342)). It would probably be fairer to Trajectron++ to adapt these helpers (where possible) to Argoverse instead of changing Trajectron's input and dynamic model.

**REVIEW CONCLUSION**

Overall, I believe the work analyzes an important problem (context integration) and takes an important direction (loss-based penalizing of context violations). At this stage, the work makes a much stronger contribution claim than its evaluations would allow substantiating. They also seem to not focus on comparing different means of context integration, which is the main area of contribution of this work. Finally, it is not clear which negative side-effects are induced by the unlikelihood loss (does the predicted variance become inconsistent? is the resulting predictor too strongly biased against "rightful" context violations?).  That being said, I think the general approach is worthy of further research and I want to encourage the authors to follow this direction.


**POST-DISCUSSION UPDATE**

I want to thank the authors for responding to my questions, correcting my misunderstandings, and addressing some of the raised points. Overall, I still believe that the work is not yet ready to be published. One of my main concerns is that a method for context integration should be evaluated in comparison to multiple other methods for context integration (ideally on multiple predictors) in order to see which approach is particularly meaningful and why.  As mentioned in the initial review, the general idea is interesting and worthy of resubmission after the authors address the issues raised in the reviews.

---

> ### Author Response · Authors · 2020-11-22
> **Answers to Questions Part 1**
>
> We thank Reviewer 3 for the detailed reviews, questions, and suggestions. We here address them and incorporated the feedback.
>
> ***The authors write "When the prediction is not so accurate (e.g. at the beginning of training), our ground truth may be farther than the negative location. In this case,
> ) will expand to better cover the ground truth and make the prediction more uncertain". I am not sure I fully understand why this is happening.***
>
> This statement comes from Eq.6. The sign of Eq.6 is determined by the distance between the predicted center and ground truth $||y_{gt,t}  - \hat \mu_t||$ and the distance between the negative point and the predicted center $||y_{neg,t}  - \hat \mu_t||$. In case the predicted center $\hat \mu_t$ is more close to the ground truth $y_{gt,t}$, Eq.6 is positive and the uncertainty of prediction will be reduced. However, if the predicted center is more close to the negative point $y_{neg,t}$, Eq.6 is negative and increase the uncertainty of the prediction. This is a usual case at the beginning of the training due to the poor performance of an untrained model.
>
> ***It is also claimed that the "coefficient gamma in Alg.1 increases gradually from 0 to 1 as a sigmoid function centered at 24th epoch." Is this rescaled in some way to ensure the graduality?***
>
> Yes. We use this method to gradually turn on the unlikelihood loss during training as we describe in the last question that this loss is not so suitable at the beginning of training.
>
> ***When describing FDE-1, the authors write "In both original Trajectron++ and our approach, this single trajectory is drawn from predicted distribution by greedy search step by step". What does the search look for? The closest sampled trajectory? If yes, why not simply use the mean?***
>
> By greedy search, we pick up the location that has the highest likelihood in our predicted distribution.
> This is the mean in case we use Gaussian distribution to model the predicted distribution. In experiments, both our method and Trajectron++ using Gaussian mixture model. In this case, we first select the Gaussian component that has the highest probability. Then, the mean of this component is used.
>
>
> ***The claim "a distribution that allocates reasonable probability mass to the region where Yi is located by minimizing the KL-divergence of predicted distribution and ground truth distribution." is in its generality not true. MLE does not require the use of KL-Divergences. This only happens in some models, particularly those that use (ammortized) variational inference such as VAEs.***
>
> Assume the ground truth data $y_{gt}$ is sampled from a data distribution $p(y)$ and the predicted distribution is $q(y)$. The KL divergence between $p(y)$ and $q(y)$ is $KL(p(y)||q(y)) = E_{y_{gt} \sim p(y)}[log \frac{p(y_{gt})}{q(y_{gt})}] = E_{y_{gt} \sim p(y)}[log p(y_{gt})] - E_{y_{gt} \sim p(y)}[log q(y_{gt})]$. Note that the first term is the entropy of the data distribution, which is not related to the model (and is constant in case the data distribution doesn't change). And the second term is the MLE loss. Therefore, maximizing likelihood of ground truth data is equivalent to minimizing the KL-Divergence between the real data distribution and predicted distribution. This statement is general.
>
> ***Shouldn't the split be between different locations? I.e., are all vehicles in NuScene's test set from different locations as in the training set? ***
>
> As nuScenes and Argoverse are recorded in the real world, different vehicles in different scenes have different locations and different orientations, which leads to different context information for each data. Therefore, all vehicles in the test set are from different locations as in the training set for both nuScenes and Argoverse.

---

> > ### Comment · AnonReviewer3 · 2020-11-22
> > **Follow-Up on Answers**
> >
> > Thank you for the answers and the clarifications!
> >
> > > Assume the ground truth data $y_{gt}$  is sampled from a data distribution $p(y)$ and the predicted distribution is $q(y)$. The KL divergence between  and  is $KL(p(y)||q(y)) = E_{y_{gt} \sim p(y)}[log \frac{p(y_{gt})}{q(y_{gt})}] = E_{y_{gt} \sim p(y)}[log p(y_{gt})] - E_{y_{gt} \sim p(y)}[log q(y_{gt})]$. [...]
> >
> > My apologies, you are absolutely right about this. I initially misunderstood.
> >
> > > As nuScenes and Argoverse are recorded in the real world, different vehicles in different scenes have different locations and different orientations, which leads to different context information for each data. Therefore, all vehicles in the test set are from different locations as in the training set for both nuScenes and Argoverse.
> >
> > My understanding is that agents in nuScenes traverse some of the locations several times. Thus my worry is that there might be some overfitting to a location even if two agents do not pass through exactly the same position. Not sure if this overfitting is happening in practice (because there are so many different locations in the dataset).

---

> > > ### Author Response · Authors · 2020-11-24
> > > **Follow-Up**
> > >
> > > Thank you for your response! We use the official validation set as our test set for nuScenes. The split is not based on location. However, the context that the model receives is centered on the position of the agent and rotated according to the orientation of the agent. Therefore, if two agents do not pass through exactly the same position, even they are in the same street, their received contexts are different. Therefore, our model needs to learn to generalize instead of simply remembering the trained context. We think this is more related to the design of nuScenes and Argoverse. If this is an issue, this will be an issue for all methods that use context information and evaluated in nuScenes and Argoverse.

---

> ### Author Response · Authors · 2020-11-23
> **Answers to Questions Part 2**
>
> ***The evaluation does not sufficiently support the author's claim that the proposed method is generally useful (and to what extent) beyond Trajectron++.***
>
> The overall improvement is about 3-5 \% when our method is applied.
> Our method focuses more on better utilizing the context information to improve the prediction quality by making the prediction follows context more. As metrics ADE/FDE are based on the geometric distance only, they cannot directly reflect the performance in the context. Therefore, we additionally evaluate our method by the context violation rate measured by our context checker and updated our experiment section to include these results (see Table 1 in the revised paper).  Our method helps reduce the violation rate from 10.6\% to 8.9\%. About 16\% context-violated prediction is avoided by our method.
>
> ***The unlikelihood loss term itself and adding the epsilon term within the log seem both to have no theoretical justification***
>
> The overall idea of unlikelihood loss is to encourage the model to allocate less probability density to the region that violates the contextual information.
> In our paper, this is done by minimizing the likelihood of trajectories sampled from such a region directly. In addition, the gradient analysis in section 3.5 shows that our loss pushes the predicted center away from the negative region to avoid context-violated prediction in case prediction is modeled by a simple Gaussian distribution. We revised the paper to extend the gradient analysis to the Gaussian Mixture Model in Appendix B of the revised paper. The conclusion in the simple Gaussian distribution case holds for Gaussian Mixture Model. Our loss pushes the center of each Gaussian component in GMM away from the negative location.
>
> The epsilon term is used for numerical stability. Epsilon will not change the monotonicity of the unlikelihood loss. By minimizing the unlikelihood loss with the epsilon, we still minimize the likelihood of trajectories that violate the context. Similar tricks are widely used (e.g., in BatchNorm).
> Mathematically, $\frac{d log(x + \epsilon)}{d x} = \frac{1}{x+\epsilon} = \frac{x}{x+\epsilon} \frac{1}{x} = \frac{x}{x+\epsilon} \frac{d log(x)}{d x}$. Epsilon here works to scale the gradient in case x is not much greater than epsilon.
>
> ***How was epsilon from eq (3) chosen? ***
> We ran a sequence of experiments for different epsilon from 1, 1e-1, 1e-2 to 1e-9 and 0 on nuScenes. The training with 0 was unstable. The best performance on the validation set is from epsilon 1e-8 and 1e-9. Since 1e-9 is smaller than 1e-8, we select 1e-9 as our epsilon. In general, smaller epsilon performs better than bigger epsilon.
>
>
>
> ***The new loss term might induce an undesired bias for scenarios where violating context would be the right thing to do and the reasonable thing to predict. ***
>
> As we mention in the paragraph ''Design of Our Chcker'' on page 4, in case the ground truth violates the checker, we don't apply our unlikelihood loss. We use this approach to alleviate this issue.
>
>
> ***I am not sure if the final loss (4) proposed in this work gives rise to any meaningful probabilistic interpretation.***
> The direct probabilistic interpretation of our unlikelihood loss is that we reduce the likelihood of context-violated trajectories in our predicted distribution to reduce the probability density in the context-violated region. Therefore, the predicted distribution is trained to obey contextual information more. In addition, we can view the predicted distribution as a policy in RL setting, and the sampled trajectories as actions. In this case, Eq.4 gives a +1 reward to the ground truth action and -$\lambda$ reward to the negative action.
>
>
> ***The authors claim that "Eq.4 is also easily adapted to combine with any other models". This is not necessarily true.***
> Thank you for your pointer. Yes, we need models that predict trajectory distributions as output.  We have revised this part for a more accurate statement.
>
>
> ***In practice the prediction horizon can be known at training time and predictors do not need to generalize to other horizons***
>
> We think a longer prediction horizon can be viewed as an evaluation way to better demonstrate the performance.
>
>
> ***Could you mention how many iterations there are per epoch. This would simplify understanding the extent of learning rate decay.***
>
> For nuScenes each epoch contains 3639 iterations. For Argoverse is 765.
>
> ***It would probably be fairer to Trajectron++ to adapt these helpers (where possible) to Argoverse instead of changing Trajectron's input and dynamic model.***
>
> I think our setting is fair. Both methods use the same input, the same dynamic model, and the same hyperparameter setting. The improvement can be only made by the proposed unlikelihood training loss. In addition, nuScenes offers the ground truth orientation for vehicles, which is needed by the Trajectron++'s dynamic model but not offered by Argoverse dataset.

---

> ### Author Response · Authors · 2020-11-24
> **Answers to Questions Part 3**
>
> ***Would it be possible to train Trajectron in a way that does not involve context-input as part of the ablation studies?***
>
> We added this ablation study in Appendix C of the latest version. Here, we remove the map input to both Trajectron++ and our model. We notice that 1) Our method without the map input (1.42) outperforms Trajectron++ without map input (1.52) measured by the FDE-Full in the 3-second case and even achieves a comparable result compared with full Trajectron++ (1.46). This phenomenon verifies that our method can inject context information into the learning signals. 2) The contribution of the map input in Trajectron++ is 6 cm (from 1.52 to 1.46). However, when we further add our unlikelihood loss, we obtain a 12 cm improvement (from 1.46 to 1.34). Our method triples the contribution of the context information.

---

### Official Review · AnonReviewer2 · 2020-10-29
**Useful Concept Beyond Motion Forecasting. Quantitative Improvements Small and needs more Theoretical Justification**

**Rating:** 6
**Confidence:** 3

**Review:**

In this paper, the authors focus on vehicular motion forecasting on roadways. To this end, they propose an interesting tweak to existing approaches. In addition to maximizing the likelihood of ground truth trajectories, the authors consider an "unlikelihood" weighted subloss which penalizes sections of the event space that shouldn't happen with context (such a driving on the wrong side of the road). They do this by sampling trajectories and labeling them with a context checker. They evaluate their approach qualitatively and quantitatively on the Argoverse and nuScenes datasets. They show improved quantitative performance over baselines.

Pros:

1. The authors focus on a problem with immediate societal applications. Motion forecasting is important for autonomous vehicles and robotics in general.

2. The paper is well written and easy to understand.

3. The proposed approach (unlikelihood training) makes intuitive sense and is useful for any situation with negative examples or known prior boundaries on the event space.

Cons:

1. Quantitative improvement over baselines is fairly small on an absolute level.

2. It would be useful to have more mathematical/theoretical justification for unlikelihood loss. I am certain that estimating a probability distribution subject to boundaries on the event space has been encountered many times before in machine learning and statistics. Could this problem be considered an optimization problem with constaints?

Overall, the paper focuses on a timely and useful problem of vehicular motion prediction, but stronger quantitative results and more theoretical grounding would be ideal.

---

> ### Author Response · Authors · 2020-11-23
> **Answers to Questions**
>
> We thank Reviewer 2 for the valuable and helpful comments. We incorporated the feedback here.
>
> ***Quantitative improvement over baselines is fairly small on an absolute level***
>
> The overall improvement is about 3-5 \% when our method is applied.
> Our method focuses more on better utilizing the context information to improve the prediction quality by making the prediction follows context more. As metrics ADE/FDE are based on the geometric distance only, they cannot directly reflect the performance in the context.
> Therefore, we additionally evaluate our method by the context violation rate measured by our context checker and updated our experiment section to include these results (see Table 1 in the revised paper).
> Our method helps reduce the violation rate from 10.6\% to 8.9\%. About 16\% context-violated prediction is avoided by our method.
>
>
> ***It would be useful to have more mathematical/theoretical justification for unlikelihood loss.***
>
> Thank you for your suggestions! We additionally analyzed the effectiveness of our loss in the case of a Gaussian Mixture Model in the revised version. For each Gaussian component, our loss pushes its center away from the negative location and toward the ground truth location, similar to the simple Gaussian distribution case shown in section 3.5.

---

### Author Response · Authors · 2020-11-23
**Official Comment**

We thank reviewers for their valuable and insightful feedback. We are encouraged that they found our method useful (R2), and our writing clear and easy to understand (R2, R3), our idea cool, and provide a very strong prior (R3). We are replying to the reviewers' comments individually and are incorporating the feedback.

---

### Author Response · Authors · 2020-11-24
**Official Comment Part 2**

We revised the paper to incorporate reviewers' feedback.
The key updates are 1) We additionally evaluated our method by the context violation rate measured by our context checker and updated our experiment section to include these results in Table 1.
Our method helps reduce the violation rate from 10.6\% to 8.9\%. About 16\% context-violated prediction is avoided by our method.
2) We extend the gradient analysis of our loss for the simple Gaussian distribution case in Section 3.5 to the Gaussian Mixture Model case in Appendix B.

---

### Author Response · Authors · 2020-11-24
**Official Comment Part 3**

We added a new ablation study in Appendix C. Here, we remove the map input to both Trajectron++ and our model. We notice that 1) Our method without the map input (1.42) outperforms Trajectron++ without map input (1.52) measured by the FDE-Full in the 3-second case and even achieves a comparable result compared with full Trajectron++ (1.46). This phenomenon verifies that our method can inject context information into the learning signals. 2) The contribution of the map input in Trajectron++ is 6 cm (from 1.52 to 1.46). However, when we further add our unlikelihood loss, we obtain a 12 cm improvement (from 1.46 to 1.34). Our method triples the contribution of the context information.

---

### Decision · Program_Chairs · 2021-01-07
**Final Decision**

**Decision:**

Reject

**Comment:**

This paper makes use of the unlikelihood objective from Welleck et al (2019) which was shown in NLP to the problem of forecasting motion trajectories on roads. The unlikelihood term is meant to lower the probability mass in non-driveable areas. The paper makes use of Trajectron++, and existing trajectory forecasting model to demonstrate the idea. While the idea is interesting, the notion of using negative examples to lower the likelihood outside a valid domain has been used in multiple occasions. The paper mentions contrastive learning, but I did not see a meaningful discussion on the difference between unlikelihood training and contrastive learning, beyond what exists in the related works section. Also, due to the unlikelihood term having appeared in Welleck et al, reviewers are hesitant to acknowledge novelty of the method. One of the reviewers also questions the significance of the results, which the authors countered by saying that their method reduces the violation rate from 10.6% to 8.9% in their predictions. This is good, but combined with the former issue implies that the paper needs more work before publication.